# Evaluation of the Properties of 3D-Printed Onyx–Fiberglass Composites

**DOI:** 10.3390/ma17164140

**Published:** 2024-08-21

**Authors:** Jong-Hwan Yun, Gun-Woong Yoon, Yu-Jae Jeon, Min-Soo Kang

**Affiliations:** 1Regional Innovation Platform Project of Kongju National University, Kongju National University, Cheonan 31080, Republic of Korea; yunjh0915@kongju.ac.kr; 2Division of Smart Automotive Engineering, Sun Moon University, Asan 31460, Republic of Korea; zzrjsdnd@sunmoon.ac.kr; 3Department of Medical Rehabilitation Science, Yeoju Institute of Technology, Yeoju 12652, Republic of Korea; superlittle@yit.ac.kr

**Keywords:** 3D printing, onyx, fiberglass, composite, tensile test, 3-point bending test

## Abstract

This study evaluated the properties of 3D-printed Onyx–fiberglass composites. These composites were 3D-printed with zero, one, two, three, and four layers of fiberglass. Ten samples of each configuration were printed for the tensile and flexural tests. The average tensile strength of the Onyx specimens was calculated to be 44.79 MPa, which increased linearly by approximately 20–25 MPa with each additional fiberglass layer. The elastic moduli calculated from the micromechanics models were compared with the experimental values obtained from the tensile tests. The experimental elastic modulus increased more significantly than the model prediction when more fiberglass layers were added. The flexural modulus of Onyx was 17.6 GPa, which increased with each additional fiberglass layer. This quantitative analysis of composites fabricated using 3D printing highlights their potential for commercialization and industrial applications.

## 1. Introduction

Recently, three-dimensional (3D) printing technology has advanced rapidly, with contributions from chemists and material engineers involved in product design and manufacturing. Also known as additive manufacturing or solid freeform fabrication, 3D printing is a manufacturing process that creates objects by layering materials based on 3D model data. This technology offers various advantages over traditional manufacturing methods, such as subtractive machining. While subtractive machining involves cutting or removing materials to create a desired shape, 3D printing builds objects layer-by-layer, reducing material waste and providing significant benefits in terms of time and cost. The 3D printing technology has been highlighted in various industries [1,2,3]. For instance, in the automotive industry, it is used for the rapid prototyping of parts and the manufacturing of customized components. In the electronics industry, it is recognized for its ability to precisely create complex circuit boards and components. In the aerospace industry, it is used to produce lightweight structures and intricate engine parts, thereby contributing to improved fuel efficiency and cost reduction. The medical field also benefits significantly from 3D printing, utilizing it to create personalized implants, prosthetics, and complex surgical tools tailored to specific patient needs [4,5,6,7]. The 3D printing industry is primarily based on polymer materials with low melting points, which are widely used due to their light weight, low cost, and processing flexibility. Initially, thermoplastic polymers such as polylactic acid (PLA) and acrylonitrile butadiene styrene (ABS) were commonly used in 3D printing. These materials can easily form complex shapes because they melt and form layers at relatively low temperatures. However, these polymers have limitations in terms of their mechanical strength and heat resistance, restricting their use in high-value-added products. The tensile strength of PLA typically ranges between 50 and 60 MPa. According to a study, the average tensile strength of PLA was found to be approximately 56.6 MPa when printed under optimal conditions. Also, the tensile strength of ABS is around 40 MPa, and PETG’s is about 50 MPa. PLA is relatively strong and easy to print with, and its mechanical properties can be further optimized through a careful adjustment of the printing parameters. Its strength makes it suitable for a variety of applications, although its lower toughness and impact resistance can be a limitation for more demanding uses. To address these limitations, various polymer composites were developed by adding reinforcements such as short fibers or nanoparticles to the polymer matrix. These composites exhibited significantly enhanced mechanical strength and heat resistance compared to pure polymers. Composites mixed with carbon or glass fibers exhibited a combination of high strength and a light weight, making them suitable for use in the aerospace, automotive, and sports equipment industries. Polymer composites with nanoparticles not only have excellent mechanical properties but also exhibit enhanced electrical and thermal properties, making them particularly valuable in the electronics industry. Polymer materials are widely used in the 3D printing industry owing to their light weight, low cost, and processing flexibility. Although 3D-printed polymer products can create geometrically complex shapes, they still face significant challenges in many applications because of their low mechanical strength and functionality [8,9]. To address these challenges, various studies on enhancing the properties of composites by adding reinforcements, such as short fibers or nanoparticles, to the polymer matrix have been conducted. These studies have garnered significant attention in academia and provide potential alternatives to overcome existing limitations by developing composites reinforced with particles, fibers, or nanomaterials. Also, 3D printing, also known as additive manufacturing, uses only the necessary amount of material to create parts, significantly reducing waste compared to traditional subtractive manufacturing methods. This reduction in waste leads to less environmental impact. Additionally, 3D-printing materials are thermoplastic materials, which are generally easier to recycle. Used plastic can be melted down and reformed into new products, supporting a circular economy and reducing the need for new raw materials.

Despite the significant progress made in the application of polymer composites in 3D printing over the past few years, this technology has not been widely adopted in most industries owing to several limitations. At present, the materials available for 3D printing are limited to thermoplastic polymers with low glass-transition temperatures, powder materials, and a few photopolymers. These limited materials are difficult to use in high-temperature environments or in applications requiring high strength. Additionally, although reinforcements enhance the performance of polymer composites, most printed composites still have a lower mechanical strength than those manufactured using traditional molding methods [10]. This could be due to the non-optimized distribution and arrangement of the reinforcements during the 3D printing process. In traditional molding methods, reinforcements are uniformly distributed within the polymer matrix; however, in 3D printing, the layer-by-layer approach can lead to an uneven distribution, reducing the strength of the printed composites. Various approaches have been proposed to address this issue. For example, chemical and physical treatments to optimize the interaction between the polymer matrix and reinforcements, as well as improvements in the 3D printing process to achieve a more uniform distribution of reinforcements, have been studied. In addition, ongoing research focuses on developing new polymer materials and reinforcements that significantly enhance the performance of 3D-printed composites. In conclusion, 3D printing technology using polymer composites has great potential but still faces many challenges. Thus, developing new materials and processes, along with quantitatively evaluating the mechanical and physical properties of 3D-printed composites to obtain reliable data, is essential. These data are crucial for the widespread adoption of 3D printing technology in various industries, enabling the development of more efficient and innovative products.

To address these challenges, Markforged developed the Onyx filament, which is a high-strength thermoplastic material that combines commercial nylon with short carbon fibers, offering an excellent surface finish and high resistance to chemical agents. Onyx serves as the thermoplastic matrix for the composite components. It can be printed alone or reinforced with continuous fibers to increase their strength. Short carbon fibers mixed into nylon control the material’s performance during cooling, resulting in less thermal distortion and making them suitable for 3D-printing composites [11,12,13]. Various studies have been conducted on printing Onyx-based composites, incorporating carbon fiber and fiberglass for industrial applications [14,15,16,17,18]. These studies have explored the properties of composites based on the printing method or the conditions of the reinforcements embedded within them. However, for a quantitative analysis of these composites, it is necessary to consider the quantities of the matrix and reinforcement used. Studying the quantitative aspects of Onyx–fiberglass composites is necessary because, in 3D printing, they solidify into a filament form and then melt under heat to form structures. Additionally, the fiberglass fibers are coated with resin to facilitate adhesion and form fiber filaments. This characteristic necessitates quantitative analysis of the composites. Furthermore, it is essential to measure and establish a basic database of the quantitative properties of 3D-printed composites for industrial applications.

Therefore, in this study, we investigated the properties of 3D-printed Onyx-based fiberglass composites.

## 2. Materials and Methods

### 2.1. Materials

The properties of the materials used in this study are listed in Table 1. Onyx is a nylon-based material mixed with 20% chopped carbon fibers to form filaments. Owing to its fine carbon fiber reinforcement, Onyx is approximately twice as strong as ABS and is suitable for applications requiring a high mechanical performance. The heat deflection temperature of Onyx is 145 °C, providing excellent thermal stability and strong resistance to various chemicals. This material can be reinforced using various mechanisms, including continuous fibers such as carbon fibers, Kevlar, and fiberglass, to enhance its properties. Onyx, developed by Markforged, is highly useful for manufacturing high-performance parts using 3D printing technology and is widely used across various industries. Its use enables manufacturers to produce high-quality parts cost-effectively, making it a widely utilized base material for composite 3D printing.

The cross-sections of the filaments used in this study are shown in Figure 1. The cross-section of the Onyx filament in Figure 1a reveals black dots distributed inside the filament, indicating the presence of chopped carbon fibers. The cross-section of the fiberglass filament in Figure 1b shows that it is composed of resin mixed with bundles of fiberglass. The diameter of Markforged’s Onyx filament was found to be 1.75 mm, while that of the fiberglass filament was 0.35 mm, both of which are consistent with the datasheet specifications [19].

### 2.2. 3D Printing Conditions

In this study, the material was printed using a Mark Two 3D printer from Markforged (Waltham, MA, USA) to evaluate the properties of the Onyx–fiberglass composites. The printing conditions are shown in Figure 2. For the printing process, Onyx was used to construct the outer wall (four layers) of the object, and the top and bottom surfaces were printed at an angle of 45° to form the exterior and interior structures. In addition, fiberglass was printed in the longitudinal direction (0°) to ensure alignment within the Onyx matrix. The internal density of both Onyx and fiberglass was set to 100% (solid state). To analyze the effect of fiberglass within the Onyx matrix, specimens were prepared with 0–4 layers of fiberglass. The thicknesses of the fiberglass layers formed inside the Onyx matrix were 0.1 mm. To perform the 3D printing process, the nozzle temperature for Onyx was set to 275 °C, the nozzle temperature for fiberglass was set to 275 °C, and the print speed was set to 100 mm/s. In addition, the volumes of Onyx and fiberglass used in the printed composite specimens were measured, and these values are listed in Table 2. We also compared the external shapes of each specimen and examined the reproducibility of the specimens through cross-sectional observations of representative specimens. Ultimately, ten identical specimens were printed, and tensile and bending tests were conducted.

### 2.3. Mircromechanics

The rule of mixtures (ROM) is another important theoretical approach in micromechanics for predicting the mechanical properties of composite materials, such as the elastic modulus of fiber-reinforced composites [20,21,22]. The micromechanics of composites play a crucial role in optimizing and predicting their performance, allowing for a more scientific and systematic approach to the design and application of these materials. Among these analytical models, the ROM is the simplest form of the micromechanics model. It linearly combines the properties of each material based on the volume fraction of each component (fiber and matrix). The micromechanics model using the ROM assumes a perfect bonding between the fiber and the matrix, with no slippage or detachment at the interface. Additionally, when a load is applied to the composite, it is assumed that all components experience the same strain. The equation for the micromechanics model using the ROM is as follows:(1)$$E_{c}=V_{f}E_{f}+V_{m}E_{m}$$
where $E_{c}$ is the effective elastic modulus of the composite, $E_{m}$ is the elastic modulus of the matrix, $E_{f}$ is the elastic modulus of the fiber, $V_{f}$ is the volume fraction of the fiber, and $V_{m}$ is the volume fraction of the matrix.

The Halpin–Tsai model is another micromechanical model for predicting the effective elastic modulus of composite materials. This model is widely used to predict the mechanical properties of fiber-reinforced composites. It models the overall behavior of composite materials based on the interaction between the fibers and the matrix (matrix material). This model assumes a perfect bonding between the fibers and matrix without considering detachment at the interface. In addition, the composite material is assumed to be microscopically uniform, and the numerical calculations are based on this assumption.
(2)$$E_{c}=E_{m}\left(\frac{1+\xi \eta V_{f}}{1-\eta V_{f}}\right)$$
where $E_{c}$ is the effective elastic modulus of the composite, $E_{m}$ is the elastic modulus of the matrix, $E_{f}$ is the elastic modulus of the fiber, and $V_{f}$ is the volume fraction of the fiber. Additionally, η is a parameter that represents the difference in properties between the fiber and the matrix, and it is defined as follows:(3)$$\eta =E_{m}\left(\frac{E_{f}-E_{m}}{{E_{f}+\xi E}_{m}}\right)$$
where ξ is a parameter that varies depending on the fiber’s shape and orientation, acting as an important variable in composite material design. For short fibers, ξ is generally 1/2, representing the effect when fibers are randomly oriented. For continuous fibers, ξ is typically 2, indicating that fibers are aligned in a specific direction, primarily in the load-bearing direction. Therefore, ξ was set to 2 in our calculations.

### 2.4. Tensile Test

The tensile testing of composite materials is crucial for evaluating their mechanical properties. In this test, the tensile strength, elastic modulus, and stress–strain behavior of the composites are measured to understand their overall performance. Because composites are fabricated by combining various materials, understanding the characteristics of each component and their interactions is important. Therefore, tensile testing is essential for assessing the durability and applicability of composite materials. Tensile tests were conducted according to the ISO 527 standard using a Tinius Olsen (Horsham, PA, USA) 100ST machine, as shown in Figure 3. ISO 527 is used to evaluate the tensile properties of the rigid thermoplastics and thermosetting plastics, fiber-reinforced plastics, as well as films and sheets made from these materials [23,24]. This standard is applicable to plastic materials of various forms and properties. The test was conducted by applying a tensile force at a speed of 5 mm/min, and the elongation of the gauge length of 50 mm was measured in real time using an extensometer, as shown in Figure 3. For each set of 3D-printed specimens, ten samples were prepared and tested to ensure the statistical significance and consistency of the test results. The repeated testing facilitated the calculation of average values and standard deviations, enabling a comparative analysis of the mechanical properties of the composites under various printing conditions.

### 2.5. Bending Test

Flexural testing is widely used to evaluate the flexural strength, elastic modulus, and toughness of materials. This test measures the behavior of materials subjected to bending loads and is essential for understanding their deformation and failure characteristics. Particularly for composite materials, flexural testing plays a crucial role in assessing the interactions between various components and the complex mechanical properties of the material. Three-point flexural tests were conducted according to the ISO 14125 standard using a Tinius Olsen(Horsham, PA, USA) universal testing machine, as shown in Figure 3. ISO 14125 is an international standard for evaluating the flexural properties of fiber-reinforced plastic composites [25,26]. Additionally, this standard provides detailed methods for measuring the flexural strength, elastic modulus, and other flexural-related properties of the composites. The flexural stress was applied at a loading rate of 5 mm/min, and the material’s deformation was measured and analyzed until a significant change was observed. For each set of 3D-printed composite material specimens, 10 samples were prepared and tested to ensure consistency and statistical significance. The repeated tests allowed for a comparative analysis of the flexural properties of the composites under different printing conditions.

## 3. Results and Discussion

### 3.1. Cross-Sectional Analysis of Specimens

To verify the printing quality of the composite materials, tensile test specimens were sectioned, and their cross-sections were examined at 20× magnification, as shown in Figure 4. The specimen with zero layers of fiberglass in Figure 4a contained only Onyx distributed in the middle section where the fiberglass should have been printed. The infill density was set to 100% (solid) during the 3D printing process, resulting in the Onyx material fully occupying the interior of the printed parts. The specimen with one layer of fiberglass, as shown in Figure 4b, featured a single layer of fiberglass printed at the center, with a thickness of 0.1 mm. The specimen in Figure 4c featured two layers of fiberglass printed symmetrically in the center of the specimen, and the gap between the two layers was 0.3 mm. The observations of the 3D-printed specimens shown in Figure 4 confirm that Onyx served as the matrix material, with the fiberglass symmetrically arranged around the center. The gap between the fiberglass layers was measured to be 0.3 mm.

### 3.2. Analysis of Tensile Test Results

After the tensile tests, the specimens closest to the average value were selected as representative samples, and their shapes are illustrated in Figure 5. Under the test conditions, fractures occurred within the gauge length, thereby validating the tensile tests. The gauge length is a critical factor in tensile testing, as it ensures an accurate measurement of strain. Accurate gauge length settings and the use of extensometers allow for the precise evaluation of the mechanical properties of the material. This evaluation provides crucial baseline data for the design, development, and application of materials, optimizing their performance in various industries. In conclusion, tensile testing and gauge length measurements are essential for evaluating the mechanical properties of materials. Accurate strain measurements are critical to ensure the reliability and reproducibility of test results [27,28,29]. Therefore, an analysis based on tensile test specimens that fractured within the gauge length was conducted to ensure the validity of the test results.

A 20×-magnification enlarged view of the fracture surface after the tensile testing is shown in Figure 6. For the specimen composed of Onyx shown in Figure 6a, a diagonal external printing condition can be observed. Additionally, the fracture surface is relatively clean owing to the absence of fiberglass inside. By contrast, the specimens shown in Figure 6b–e reveal the presence of fiberglass within the structure, with the complete fractures of the fiberglass visible. Specimens with fiberglass aligned within the structure exhibit more fiberglass protrusion than the fractured Onyx surface. This is because the elongation of fiberglass is lower than that of Onyx, causing the fiberglass to break first. This effect resulted in a fracture morphology, as shown in Figure 6.

Figure 7 shows the tensile strength of the specimens with different layers of fiberglass layers. The tensile strength of the specimen composed solely of Onyx was 44.79 MPa, while that of the specimen with one layer of fiberglass was measured to be 69.45 MPa. As the number of layers increased to two, three, and four, the measured tensile strengths increased to 97.87 MPa, 125.96 MPa, and 146.33 MPa, respectively. The tensile strength increased linearly by 20–25 MPa with each additional fiberglass layer. However, the reason for limiting the number of fiberglass layers to four is because the increase in the number of fiberglass layers makes the effect of the fibers increasingly significant, leading to inconsistent results in the tensile tests. In addition, failures occurred in the grip section without the formation of fractures within the gauge length of the tensile test specimens, thereby preventing a proper quantitative analysis.

The stress–strain curves of the specimens with different layers of fiberglass, obtained through tensile testing, are shown in Figure 8. For the tensile test specimen containing only Onyx, the strain increased by up to 20%, demonstrating characteristics similar to those of polymer materials with a low tensile strength and high elongation. For the specimens containing one and two layers of fiberglass, the tensile strength decreased after reaching the maximum level while the elongation increased. This can be attributed to the lower elongation of the fiberglass layers within the specimen, which caused the fiberglass to fracture first under tensile stress, followed by the eventual failure of the external Onyx matrix. For specimens with three and four layers, the internal fiberglass endured a higher cccctensile stress, and when subjected to excessive force, the external Onyx broke abruptly. Previous studies have reported a tensile strain at break for fiberglass to be approximately 3.8, which is consistent with the tensile test results observed in this study [30,31,32].

### 3.3. Comparison through Micromechanics

Figure 9 shows a comparison of the elastic moduli of the composites obtained from tensile testing. The elastic moduli of composite materials can be derived from experimental values or calculated using the rule of mixtures (ROM) and Halpin–Tsai models. Because both these models are used to calculate the effective properties of materials based on their volume fractions, the elastic modulus of the specimens composed solely of Onyx was calculated to be 2400 MPa, which is consistent with the basic material properties. However, the experimental results measured a pressure of 2546 MPa. As the amount of fiberglass within the composite increased, the experimentally measured elastic moduli became significantly higher than those predicted by the ROM and Halpin–Tsai models. Conversely, both models exhibited a similar trend of increasing elastic modulus with the increase in the number of fiberglass layers. This indicates that as more fiberglass layers are added, the discrepancy between the calculated values and the experimental results is expected to increase. This is because the surface bonding strength of the two materials plays a role when the composite material is printed through 3D printing. Additionally, fiberglass contains resin, which acts as an adhesive to enhance the actual bonding strength. However, since this surface adhesive effect is not considered in the analytical model, discrepancies can be observed.

### 3.4. Analysis of Flexural Test Results

The flexural properties of the composite materials were analyzed using a three-point flexural test. The flexural modulus measured from the flexural test was as shown in Figure 10. The flexural modulus of Onyx alone was 17.6 GPa. When one layer of fiberglass was oriented within Onyx, the flexural modulus increased to 24.0 GPa. With two layers, the modulus increased to 26.9 GPa, and with three layers, 32.7 GPa. Finally, with four layers of fiberglass, the modulus reached 42.5 GPa. As the number of fiberglass layers increased, the flexural modulus increased, indicating that a greater force was required for bending. This characteristic can be attributed to the higher flexural modulus of fiberglass than that of Onyx. The more layers of fiberglass there are, the stiffer the material becomes. The increase in the modulus was also due to the orientation of the fibers. The fibers aligned along the length of the sample helped distribute the stress from the center of the sample, where the load was applied, to the support points in the three-point bending test.

The load–displacement curves obtained from the three-point bending test are shown in Figure 11. The y-axis values were calculated based on the displacement recorded by the testing machine during the flexural test, and a graph was generated by measuring the real-time displacement. The maximum displacement point was set at a bending strain of 1.5. For the specimens containing zero, one, and two fiberglass layers, the bending test was continued up to the maximum displacement. However, for specimens with four and four layers of fiberglass, the test was interrupted by damage before the maximum displacement was reached. Upon examining the specimen images after the bending test, as shown in Figure 12, no external damage was observed. It is believed that the lack of visible damage is due to the higher strain of Onyx printed externally via 3D printing compared with fiberglass. Moreover, for specimens with three and four layers of fiberglass, it was inferred that the internal fiberglass was damaged, leading to the early termination of the bending test. Thus, the internal fiberglass orientation enhances the bending stress and flexural modulus, thereby providing a higher resistance to deformation.

The results of the internal examination of the flexural test specimens observed through destructive testing are shown in Figure 13. The images were analyzed at 40× magnification. For the specimens containing zero, one, and two fiberglass layers, no damage was observed, as shown in Figure 13a. However, for specimens with three and four layers of fiberglass, internal damage and cracks were observed, as shown in Figure 13b. Therefore, even in the absence of external damage, internal damage to the fiberglass can degrade the mechanical properties of the composite material.

## 4. Conclusions

In this study, tensile and flexural tests were performed to evaluate the properties of 3D-printed Onyx–fiberglass composites. The results were as follows:A cross-sectional observation of the tensile test specimens confirmed the reproducibility of the 3D printing results, with Onyx filling at a 100% infill density and 0.1 mm thick fiberglass layers symmetrically arranged at 0.3 mm intervals in the center.Tensile tests ensured validity with fractures forming within the gauge length, and an accurate mechanical property evaluation was achieved through proper gauge length settings and extensometer use.The analysis of the tensile strength changes after the tensile test showed that the average tensile strength of the Onyx specimens was 44.79 MPa. The tensile strength increased linearly by approximately 20–25 MPa with each additional fiberglass layer, reaching up to 146.33 MPa. However, beyond four layers of fiberglass, the consistency of the test results decreased, and fractures did not form within the gauge length, which complicated the quantitative analysis.The analysis of the stress–strain curve revealed that the specimens made solely of Onyx exhibited 20% strain. Specimens with one and two layers of fiberglass showed increased strain and decreased strength after reaching the maximum tensile strength. Specimens with three and four fiberglass layers demonstrated a tendency for the internal fiberglass to withstand a high tensile stress with sudden external Onyx fracturing.A comparison of the experimental values with those calculated using the ROM and Halpin–Tsai models showed that the experimental elastic modulus increased significantly more than the model predictions as more fiberglass layers were added to the composite.The results of the three-point flexural test indicated that the flexural modulus of Onyx was 17.6 GPa. The flexural modulus increased with the addition of fiberglass layers, measured at 24.0 GPa for one layer, 26.9 GPa for two layers, 32.7 GPa for three layers, and 42.5 GPa for four layers.Although no external damage to the Onyx was observed in the flexural test, internal damage to the fiberglass could degrade its mechanical properties.

Therefore, it is necessary to build a database of 3D-printed composite materials through quantitative and comparative analyses using simulations. In addition, to analyze the composite materials, it is necessary to improve the existing micromechanics model by analyzing the surface bonding strength and surface conditions between the materials and proposing quantified coefficients. Ongoing research based on this database is essential for the commercialization and industrial application of composite materials fabricated using 3D printing.

## Figures and Tables

**Figure 1 materials-17-04140-f001:**
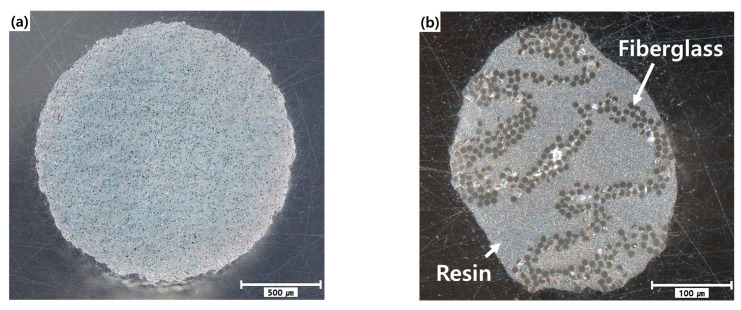
Cross-sectional shapes of the filaments. (**a**) Cross-section of Onyx filament. (**b**) Cross-section of fiberglass filament.

**Figure 2 materials-17-04140-f002:**
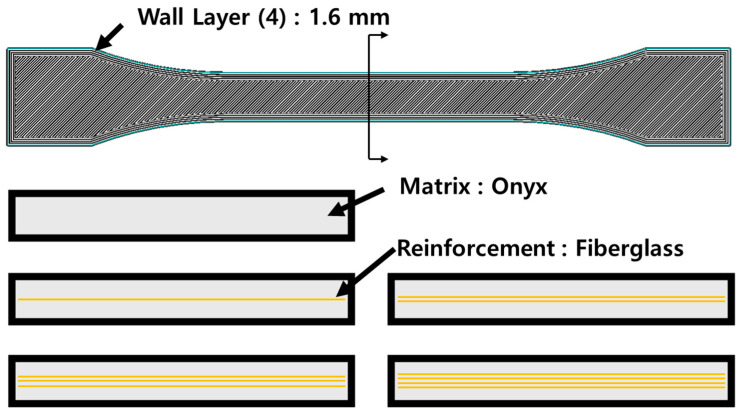
Schematic of tensile test specimen printed by 3D printing.

**Figure 3 materials-17-04140-f003:**
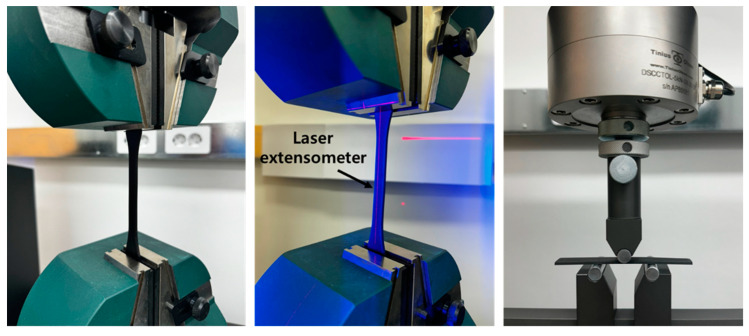
Tensile and flexural testing.

**Figure 4 materials-17-04140-f004:**
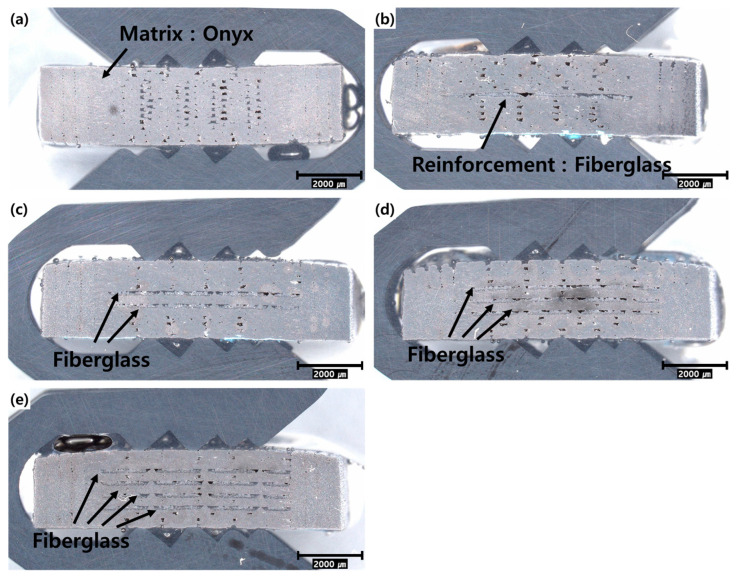
Cross-sectional images of test specimens with (**a**) 0, (**b**) 1, (**c**) 2, (**d**) 3, and (**e**) 4 layers of fiberglass.

**Figure 5 materials-17-04140-f005:**
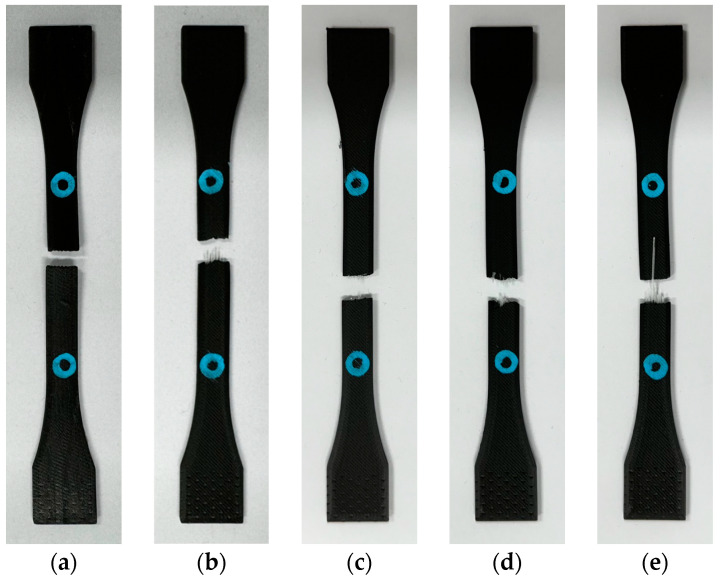
Shapes of the specimens containing (**a**–**e**) 0–4 layers of fiberglass, respectively, as observed after tensile testing.

**Figure 6 materials-17-04140-f006:**
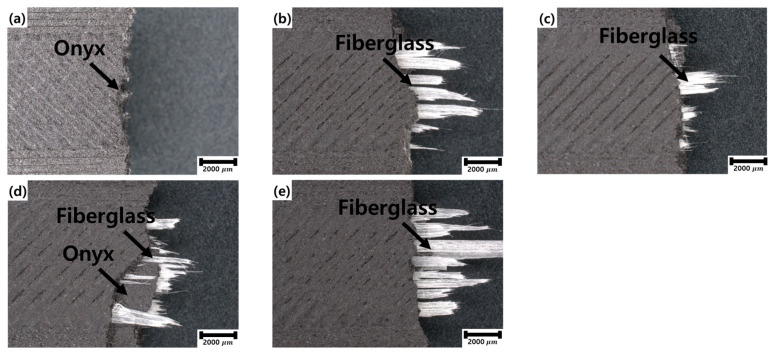
Enlarged cross-sectional view of the tensile test specimens with (**a**–**e**) 0–4 layers of fiberglass, respectively.

**Figure 7 materials-17-04140-f007:**
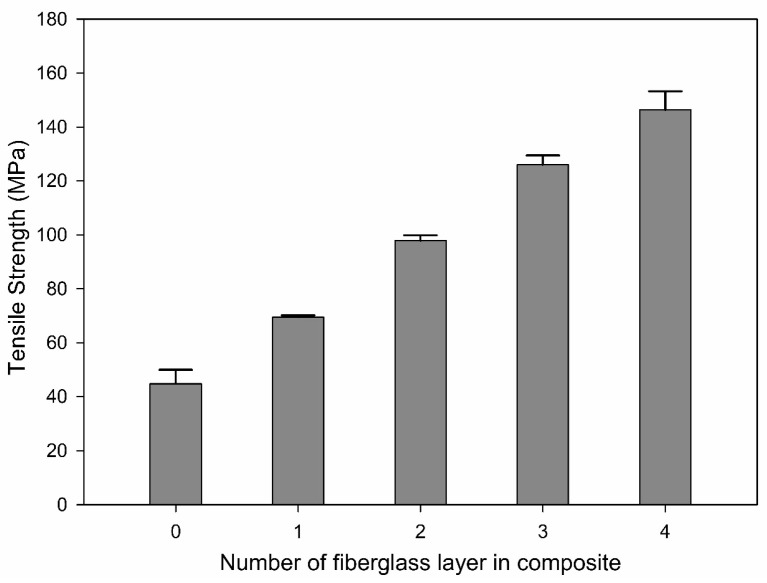
Tensile strengths of the Onyx–fiberglass composites with different layers of fiberglass.

**Figure 8 materials-17-04140-f008:**
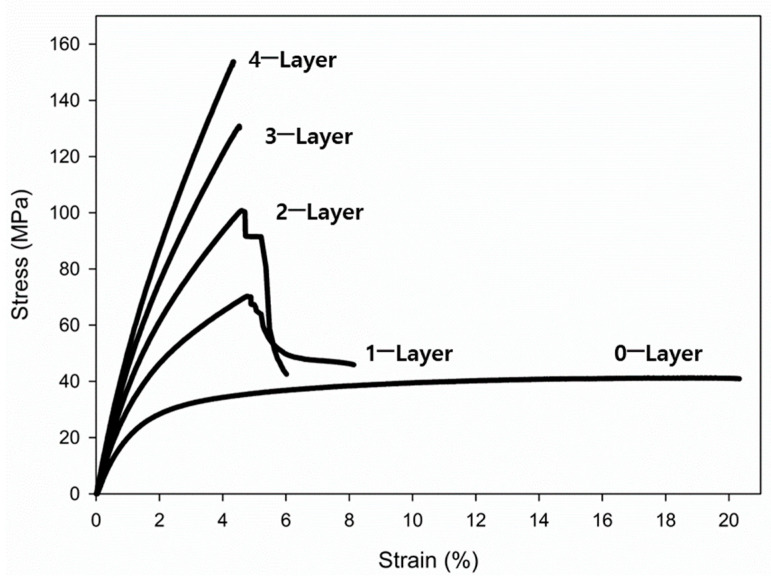
Stress–strain curves of the specimens with different layers of fiberglass. These results were obtained from tensile testing.

**Figure 9 materials-17-04140-f009:**
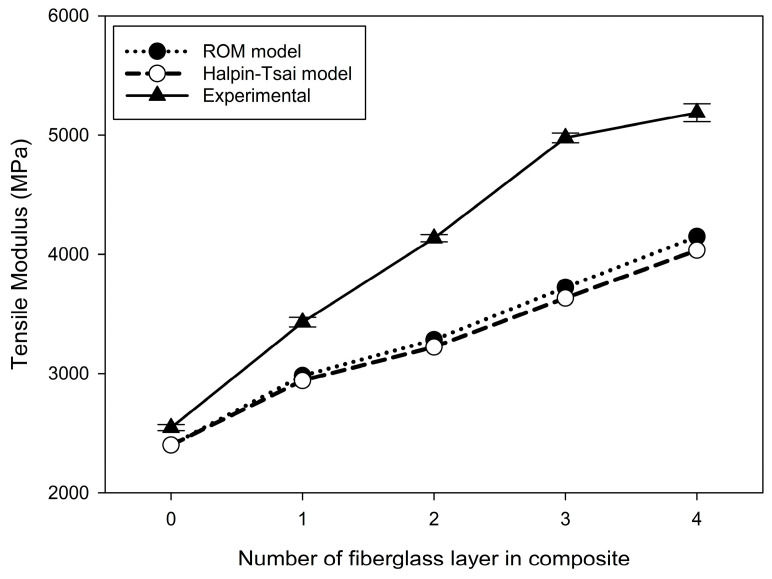
Variations in the tensile modulus of Onyx–fiberglass composite with respect to the number of fiberglass layers.

**Figure 10 materials-17-04140-f010:**
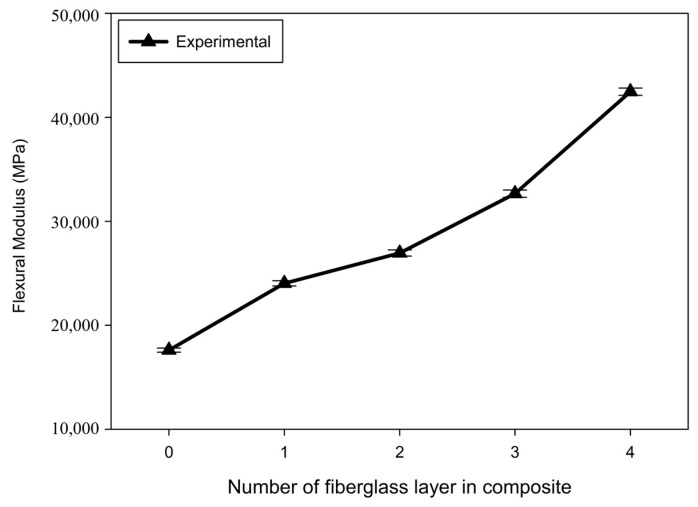
Variations in the bending modulus of the Onyx–fiberglass composite with respect to the number of fiberglass layers.

**Figure 11 materials-17-04140-f011:**
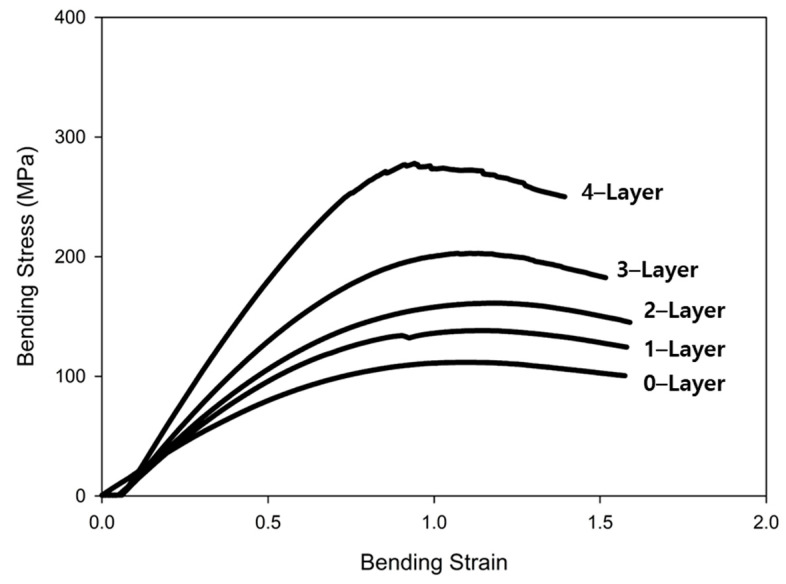
Flexural test results of the specimens with different fiberglass layers.

**Figure 12 materials-17-04140-f012:**
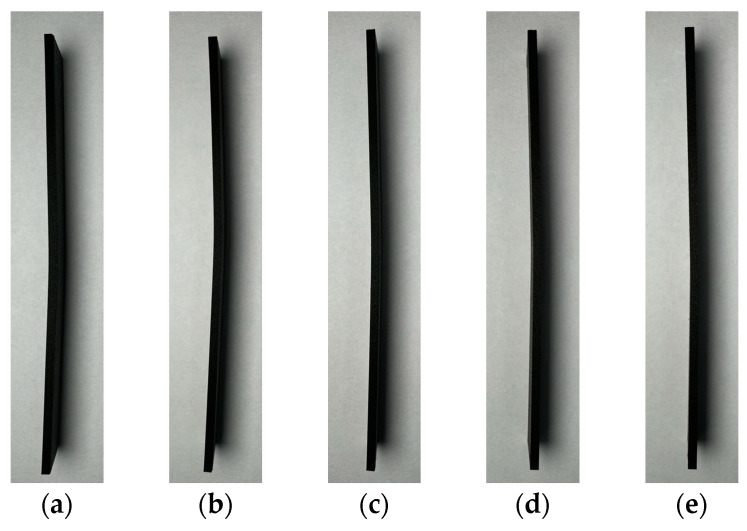
Images of flexural test specimens with (**a**–**e**) 0–4 layers of fiberglass, respectively.

**Figure 13 materials-17-04140-f013:**
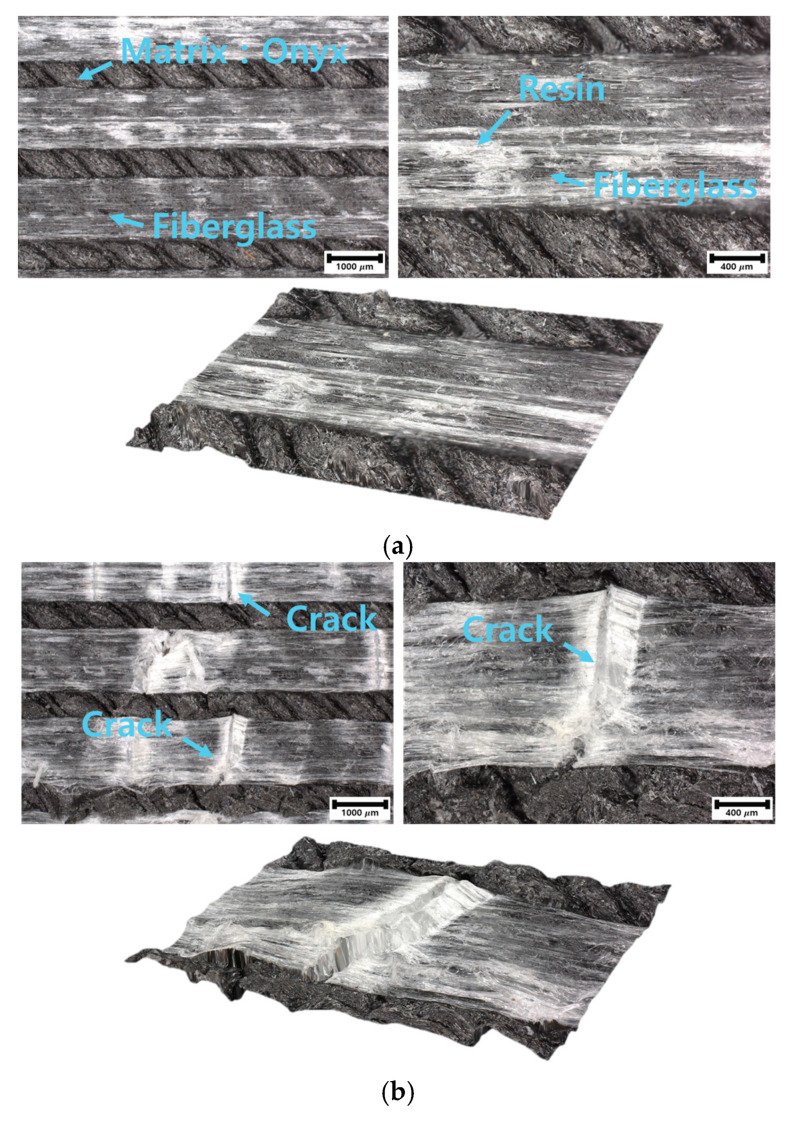
Images of fiberglass damage post flexural test. (**a**) Undamaged fiberglass section. (**b**) Broken fiberglass section.

**Table 1 materials-17-04140-t001:** Properties of Onyx and fiberglass.

Property	Onyx	Fiberglass
Tensile Modulus (MPa)	2400	21,000
Yield Tensile Strength (MPa)	41	590
Tensile Strength at Break (MPa)	37	-
Tensile Strain at Break (%)	25	3.8
Flexural Strength (MPa)	71	200
Flexural Modulus (MPa)	3000	22,000
Heat Deflection Temperature (°C)	145	105
Density (g/cm^3^)	1.2	1.5

**Table 2 materials-17-04140-t002:** Material volume fractions of the 3D-printed Onyx–fiberglass system.

Category	Materials	0–Layer	1–Layer	2–Layer	3–Layer	4–Layer
Volume	Onyx (cm^3^)	5.25	5.27	5.22	5.10	5.01
	Fiberglass (cm^3^)	0	0.17	0.26	0.39	0.52
Volume fraction	Onyx	1	0.969	0.953	0.929	0.906
	Fiberglass	0	0.031	0.047	0.071	0.094

## Data Availability

Restrictions apply to the datasets.

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
