# Peer review of "Evaluation of the Properties of 3D-Printed Onyx–Fiberglass Composites"

_materials, 2024, doi:10.3390/ma17164140_

Round 1

Reviewer 1 Report

Comments and Suggestions for Authors

The paper presents a novel exploration into the use of Onyx combined with glass fibers for 3D printed composites. This approach is innovative as it leverages the strengths of both materials to enhance mechanical properties, which is crucial for expanding the applications of 3D printing technology in various industries. By addressing the following recommendations, the paper can provide a more holistic view of the potential and challenges associated with Onyx/glass fiber 3D printed composites, making it a more valuable resource for researchers and industry professionals.

1.     Can the authors provide more details on the specific parameters used for the 3D printing process, such as temperature settings, printing speed, and layer height? How did you ensure consistency and reproducibility across the 10 samples printed for each condition?

2.     The authors mentioned that the elastic modulus calculated from micromechanics models was compared with experimental values. Could they elaborate on the discrepancies observed and potential reasons for these differences?

3.     How did the addition of fiberglass layers influence other mechanical properties such as impact resistance or hardness?

4.     The tensile strength increased linearly with the addition of fiberglass layers. Did the authors observe any saturation point or diminishing returns at higher fiber content? Could they discuss the statistical significance of their results, particularly the variation in tensile and flexural strengths with different numbers of fiberglass layers?

5.     The paper references the Rule of Mixtures and Halpin-Tsai models. How accurate were these models in predicting the composite properties, and were any adjustments needed to fit the experimental data? Did they explore any other micromechanics models or simulations to support your findings?

6.     Based on their results, what specific industrial applications do they foresee for these Onyx/glass fiber composites? What are the primary limitations or challenges in scaling up the production of these composites for commercial use?

7.     What further research do they plan to undertake to optimize the properties of Onyx/glass fiber composites? Are there any other materials or reinforcements they are considering for future studies to enhance the performance of 3D printed composites?

8.     The paper lacks a comparative analysis: How do the properties of the Onyx/glass fiber composites compare to other commonly used 3D printed composites, such as those reinforced with carbon fibers? Did authors conduct any comparative studies with traditional manufacturing methods to highlight the advantages or disadvantages of your 3D printed composites?

9.     Can authors discuss the environmental impact of using Onyx and glass fiber materials in 3D printing, particularly in terms of recyclability and material waste? What are the economic implications of adopting these materials in terms of cost-effectiveness and production efficiency compared to traditional manufacturing methods?

Author Response

1.     Can the authors provide more details on the specific parameters used for the 3D printing process, such as temperature settings, printing speed, and layer height? How did you ensure consistency and reproducibility across the 10 samples printed for each condition?

The nozzle temperature and print speed for each material used in the 3D printing process were additionally described in the '2.2 3D Printing Conditions' section. Furthermore, methods for inspecting the appearance of the specimens and analyzing reproducibility through cross-sectional observations were included.

2.     The authors mentioned that the elastic modulus calculated from micromechanics models was compared with experimental values. Could they elaborate on the discrepancies observed and potential reasons for these differences?

Good point.

This explanation has been included in the '3.3 Comparison through Micromechanics' section as follows: “This is because the surface bonding strength of the two materials plays a role when the composite material is printed through 3D printing. Additionally, fiberglass contains resin, which acts as an adhesive to enhance the actual bonding strength. However, since this surface adhesive effect is not considered in the analytical model, discrepancies can be observed.”

3.     How did the addition of fiberglass layers influence other mechanical properties such as impact resistance or hardness?

Unfortunately, due to the current lack of facilities to perform impact testing, we were unable to conduct experimental analysis. We plan to address this issue and include the results in a future paper, considering the additional testing equipment we are currently setting up.

However, based on the analysis of various existing papers on polymer composites, it is observed that the impact strength increases as more fibers are added to the composite material. Therefore, it is expected that the composite materials printed using 3D printing will exhibit similar results.

4.     The tensile strength increased linearly with the addition of fiberglass layers. Did the authors observe any saturation point or diminishing returns at higher fiber content? Could they discuss the statistical significance of their results, particularly the variation in tensile and flexural strengths with different numbers of fiberglass layers?

Currently, according to the experimental results, tensile tests cannot be accurately performed when there are more than five layers of fiber. The results are unreliable because the specimens do not break within the gauge length. This issue arises because the internal fiber layers and the Onyx matrix delaminate, causing failure at the grip section of the tensile test.  Therefore, we conducted experiments only up to four layers to collect and analyze reliable data. We are currently addressing the grip issue and conducting additional experiments to improve the results.

5.     The paper references the Rule of Mixtures and Halpin-Tsai models. How accurate were these models in predicting the composite properties, and were any adjustments needed to fit the experimental data? Did they explore any other micromechanics models or simulations to support your findings?

Micromechanics simply performs mathematical calculations based on the material properties and the volume fraction used. Although these models often differ from actual analysis results, similar trends were observed in this study. These discrepancies are likely due to differences in surface bonding strength. Recognizing this surface influence, we are analyzing its impact through surface delamination tests. Our ultimate goal is to analyze this surface influence through extensive experimentation and apply it to the model.  For the next research objective, we are continuously analyzing the surface bonding influence between the materials in the composite and aim to propose an improved model.

The potential and objectives of future research were additionally mentioned in the conclusion section.

6.     Based on their results, what specific industrial applications do they foresee for these Onyx/glass fiber composites? What are the primary limitations or challenges in scaling up the production of these composites for commercial use?

In the introduction, it is mentioned that traditional 3D printing has primarily focused on prototyping. This limitation is due to the weak strength and properties of PLA materials. Therefore, 3D printed composite materials with enhanced strength could enable the production of practical, applicable parts across various fields.

7.     What further research do they plan to undertake to optimize the properties of Onyx/glass fiber composites? Are there any other materials or reinforcements they are considering for future studies to enhance the performance of 3D printed composites?

We are conducting a study to analyze the properties of electromagnetic shielding composite materials printed using Onyx ESD, which is resistant to electromagnetic fields. This study involves surface processing and the addition of internal materials. The detailed findings will be proposed in a subsequent paper.

8.     The paper lacks a comparative analysis: How do the properties of the Onyx/glass fiber composites compare to other commonly used 3D printed composites, such as those reinforced with carbon fibers? Did authors conduct any comparative studies with traditional manufacturing methods to highlight the advantages or disadvantages of your 3D printed composites?

We have elaborated on the issues of traditional 3D printing materials in the introduction section.

9.     Can authors discuss the environmental impact of using Onyx and glass fiber materials in 3D printing, particularly in terms of recyclability and material waste? What are the economic implications of adopting these materials in terms of cost-effectiveness and production efficiency compared to traditional manufacturing methods?

We have provided more detailed explanations about the issues of traditional 3D printing materials in the introduction section, along with the answer to question 8.

Reviewer 2 Report

Comments and Suggestions for Authors

Novelty of the entire paper is a concern for this reviewer. The following points should be furnished.

The study does not report any analytical investigation. Experiments are direct. There is no ANOVA. There is no DoE. There is no Taguchi Method. There is no Response Surface Method. So, the overall study is like a senior design project. For Figure 9, there is no error or precision reported. Figure 6 was not labeled.

Continuous Fiber Reinforced AM is a topic investigated by a high number or researchers. Literature also has a high number of published archival journal papers. Paper does not report anything new.

Equations are not numbered. Their derivations/meanings are poorly presented.

Core of the research findings should be reported at the conclusion. Specifics and details should be removed.

Author Response

Novelty of the entire paper is a concern for this reviewer. The following points should be furnished.

1. The study does not report any analytical investigation. Experiments are direct. There is no ANOVA. There is no DoE. There is no Taguchi Method. There is no Response Surface Method. So, the overall study is like a senior design project. For Figure 9, there is no error or precision reported. Figure 6 was not labeled.

We have revised the study to address several key issues. First, we have added error bars to Figure 9 to indicate the precision of the data. Additionally, Figure 6 has been labeled to facilitate easier identification and understanding of the content.

We did not perform separate DoE or ANOVA analyses. Concerned that the graphs might become too numerous and complex, we chose to simplify our approach. We apologize for this limitation. In future papers, we will conduct precision analysis and data analysis on the measured data to incorporate these results into our findings.

2. Continuous Fiber Reinforced AM is a topic investigated by a high number or researchers. Literature also has a high number of published archival journal papers. Paper does not report anything new.

The application of thermoplastic composites in the field of composite materials is a significant issue. This paper is not merely a study on continuous fiber reinforced additive manufacturing (AM), but rather an analysis of the properties of composites formed through 3D printing, comparing these properties with experimental results. Additionally, problems in this area were identified. The paper suggests objectives for future research, emphasizing the analysis of surface characteristics between materials.

If our future research becomes a series and connects to the ultimate goal, it could be considered highly valuable.

3. Equations are not numbered. Their derivations/meanings are poorly presented.

Firstly, we have inserted the equation numbers. Since this equation is meant to explain micromechanics, no separate derivation is required. It is an equation used to present the results calculated from the above formula.

4. Core of the research findings should be reported at the conclusion. Specifics and details should be removed.

We have removed the detailed content from the conclusion section and mentioned only the conclusion.

Reviewer 3 Report

Comments and Suggestions for Authors

1 lines136 &137, ""seemed to be an in completed sentence, "effect on what?" should be made clear.

2 Which ones are Figures 1, 2 and 3? They have not been marked correctly.

3 In figure 9, error bar should be given.

4 When discussing the cross-section morphologies, it should be described how to get the graphs,  by SEM? or by other ways? How about the magnification?

Author Response

Thank you for reviewing our paper.

We have incorporated the following revisions as per your suggestions. Thank you.

1.       lines136 &137, ""seemed to be an in completed sentence, "effect on what?" should be made clear.

We have refined and revised the English expressions. Please find attached the English proofreading certificate.

2.       Which ones are Figures 1, 2 and 3? They have not been marked correctly.

We have adjusted the size of the figures and verified the displayed forms for corrections."

3.       In figure 9, error bar should be given.

Figure 6 has been detailed with the addition of labels. Error bars have been added to Figures 9 and 10.

4.       When discussing the cross-section morphologies, it should be described how to get the graphs,  by SEM? or by other ways? How about the magnification?

The magnification is specified on the enlarged images. However, specifying magnification is not common practice because it does not provide a clear reference point. Typically, a scale bar is used to present quantified dimensions.

Unfortunately, due to the low magnification, we were unable to utilize SEM. Observations were made using an optical microscope. If detailed defects or damages are found in the future, we will conduct analyses using SEM and EDX.

Round 2

Reviewer 1 Report

Comments and Suggestions for Authors

The paper is definitively improved.

Reviewer 2 Report

Comments and Suggestions for Authors

Authors did a very good editing and enhancements.